# Finger-inspired rigid-soft hybrid tactile sensor with superior sensitivity at high frequency

Jinhui Zhang [1], Haimin Yao [2], Jiaying Mo[3,4], Songyue Chen[1], Yu Xie[1], Shenglin Ma[1], Rui Chen[1], Tao Luo [1], Weisong Ling[1], Lifeng Qin [1] ✉, Zuankai Wang [3] ✉ & Wei Zhou [1] ✉

Among kinds of flexible tactile sensors, piezoelectric tactile sensor has the advantage of fast response for dynamic force detection. However, it suffers from low sensitivity at high-frequency dynamic stimuli. Here, inspired by finger structure—rigid skeleton embedded in muscle, we report a piezoelectric tactile sensor using a rigid-soft hybrid force-transmission-layer in combination with a soft bottom substrate, which not only greatly enhances the force transmission, but also triggers a significantly magnified effect in $d_{31}$ working mode of the piezoelectric sensory layer, instead of conventional $d_{33}$ mode. Experiments show that this sensor exhibits a super-high sensitivity of 346.5 pC $N^{-1}$ (@ 30 Hz), wide bandwidth of 5–600 Hz and a linear force detection range of 0.009–4.3 N, which is ~17 times the theoretical sensitivity of $d_{33}$ mode. Furthermore, the sensor is able to detect multiple force directions with high reliability, and shows great potential in robotic dynamic tactile sensing.

With the fast development of robotic science and technology, more and more intelligent robots have been used to take the place of human beings to perform tasks in extreme or dangerous environments. For example, during the pandemic COVID-19, intelligent robots have been widely used to transport living supplies for quarantined people, which greatly avoids cross infection and spread of the pandemic. As the indispensable components of intelligent robots, flexible tactile sensors play a pivotal role in endowing the robots with human-like tactile sensing for high dexterity operation, such as grabbing, holding, and touching[1,2]. These human-like sensors can detect the intensity and modes of diverse stimuli, including pressing, tapping, and slipping[3,4]. While, for human fingers, this ability mainly owes to four functionally mechanoreceptors (slow-adapting type I, II named as SA-I, II, and fast-adapting type I, II named as FA-I, II) scattering in the skin for static (<-5 Hz) and dynamic (5–400 Hz) force detecting[5].

To achieve the perception of mechanical force sensing for intelligent robotics and wearable electronics, tactile sensors based on piezoresistive[6–9], capacitive[4,10,11], triboelectric[12,13], and piezoelectric[14–16] mechanisms are commonly used to convert tactile information into electrical signals. Among them, piezoelectric flexible tactile sensor has the advantage of fast response for dynamic force detection[3,17], thus was widely used to mimic FA-I, II in human skin. And its form often takes a layered structure composed of flexible substrates and sensory layer[8], wherein flexible substrates are used to contact stimuli directly and transfer force from the outer to inner sensory layer, or to conform to the curved surfaces of the robot's body for interacting with the environment[18]. However, compared with rigid-materials-based tactile sensors that utilize silicon, ceramics, and glass as substrate, the sensitivity and responding speed of flexible tactile sensors are generally limited by the natural viscoelasticity of elastomeric substrate because it would absorb part of mechanical energy[19,20]. Although efforts have been dedicated to designing flexible substrates with microstructures[21,22], such as improving the electrical property of sensory layer[23–26], or changing the sensory layer from 2D to 3D[27] to

[1]Department of Mechanical and Electrical Engineering, Xiamen University, Xiamen 361102, China. [2]Department of Mechanical Engineering, The Hong Kong Polytechnic University, Hung Hom, Kowloon, Hong Kong SAR, China. [3]Department of Mechanical Engineering, City University of Hong Kong, Hong Kong 999077, China. [4]Hong Kong Centre for Cerebro-Caradiovasular Health Engineering (COCHE), Hong Kong 999077, China. ✉ e-mail: liq@xmu.edu.cn; zuanwang@cityu.edu.hk; weizhou@xmu.edu.cn

improve the sensitivity, the sensitivity of piezoelectric flexible tactile sensors remains limited[3]. Moreover, existing piezoelectric tactile sensors work in $d_{33}$ mode[26-38], which are susceptible to a theoretical sensitivity below the $d_{33}$ coefficient.

Here, inspired by the finger structures of animals and humans, which consist of skeleton embedded in muscle, such as human finger (Fig. 1a), we report a super-sensitive piezoelectric tactile sensor adopting a rigid-soft hybrid force-transmission-layer in combination with a soft bottom substrate. Some researches have shown that the rigid structure can efficiently improve the flexible tactile sensor performance, e.g., report 6 presents a soft piezoresistive tactile sensor using a rigid force-post successfully realizes normal and shear forces detection. While, to the best of our knowledge, such a rigid-soft hybrid design has not been used in piezoelectric tactile senor for highly sensitive dynamic stimuli detection. We demonstrate this hybrid structure not only significantly enhances the transmission of high-frequency dynamic force, but also brings a magnified sensitivity to the piezoelectric sensory layer in a new $d_{31}$ working mode, instead of the conventional $d_{33}$ mode. As a result, this rigid-soft hybrid tactile sensor (RSHTS) exhibits a super-high sensitivity of 346.5 pC N$^{-1}$, wide bandwidth of 5–600 Hz and a broad force range of 0.009–4.3 N, which is ~17 times the theoretical limit sensitivity in $d_{33}$ mode (see Supplementary Table 1, Supporting Information for the comparison with other piezoelectric tactile sensors). We also demonstrate that the RSHTS-based robotic hand can achieve real-time detection of multiple force directions. The bioinspired tactile sensor holds much potential applications in a wide range of robotic dynamic tactile sensing for high dexterity operation.

## Results

### Concept, structure, and performance of the bioinspired rigid-soft hybrid tactile sensor

Flexible tactile sensors, owing to their soft substrates and capability of large deformation, can accommodate surfaces of diverse shapes. However, flexible tactile sensors tend to have limited sensitivity and response rate due to the higher viscoelasticity of their elastomeric substrates, which makes most of the flexible sensors fail to detect high-frequency signals. In contrast, rigid sensors can detect high-frequency dynamic stimuli with high sensitivity because of their low mechanical damping while unable to fit the complex and varying surface geometries. If one can take advantages of rigid and soft materials in a synergetic way, the resultant tactile sensor is expected to achieve both high sensitivity at high frequency and high flexibility. In nature, such a concept for sensor design seems to have been adopted in the limbs of animals and humans, which contain interior rigid skeleton and exterior soft muscle and skin tissues.

Inspired by the fingers of human (Fig. 1a), here we design a rigid-soft hybrid tactile sensor array with three layers (Fig. 1b), namely, one top dome layer, one sensory layer, and one bottom layer. For one sensory unit, the top dome-shaped layer is made of soft polymer embedded in rigid pillars, namely a rigid-soft hybrid force-transmission-layer, which mimics limb structure (skeleton embedded in muscle) to efficiently transmit external stimuli to inner structure. The sensory layer is made of piezoelectric film attached with patterned electrodes (four top electrodes and one shared bottom electrode) to form four piezoelectric capacitors, which is used to receive the stimuli from the top force-transmission-layer and convert them into electrical signals based on piezoelectric effect. This layer acts similarly as mechanoreceptors (FA-I, II) in human skin for dynamic stimuli detection. The bottom layer is made of soft material mimicking skin dermis ($E_b \leq 1$ MPa)[39,40], which is used to support the sensory layer and connect with objects such as a robotic hand. Thus, a bioinspired rigid-soft hybrid tactile sensor is formed. In terms of the rigid-soft hybrid force-transmission-layer in this design, it could greatly enhance the transmission of high-frequency dynamic force, but also induce a magnified

effect in $d_{31}$ mode of the piezoelectric sensory layer with super-high sensitivity in the aids of the soft bottom substrate. Meanwhile, with the combination of the rigid-soft hybrid dome-shaped layer and four independent piezoelectric capacitors, the proposed sensor can precisely recognize the multiple force directions in real-time by recording the change of the capacitors output charge.

To illustrate our concept of RSHTS, we fabricated a $3 \times 3$ tactile sensor array with an area of $1.6 \times 1.6$ cm$^2$. For one sensory unit, the top dome-shaped layer ($R = 2$ mm) made of polydimethylsiloxane (PDMS) and four epoxy pillars with a radius of 0.6 mm forms the rigid-soft hybrid layer; the polyvinylidene fluoride (PVDF)-made sensory layer (thickness 30 μm) is patterned with four quarter-circle-shaped Al electrodes on the upper surface (responding to the position of four pillars) and a shared bottom Al electrode on the lower surface, as top and bottom electrode of four piezoelectric capacitors; silicone is used as the bottom layer mimicking the more flexible skin dermis (details of fabrication process can be seen in "Methods" part and Supplementary Figs. 23, 24). Although the proposed tactile sensor has rigid pillars embedded in soft PDMS, the sensor array still has similar flexibility as the purely flexible tactile sensor without rigid pillars in our design (named as control sensor) shown in Fig. 1d or Supplementary Fig. 1.

Upon the force loading, the PVDF-made sensory film of RSHTS has a bending deformation making the sensor working in $d_{31}$ mode (the generated electric field is perpendicular to the stress), which is different from the existing piezoelectric tactile sensors mainly with the deformation along the thickness making them working in $d_{33}$ mode (the generated electric field is parallel to the stress), as shown in Fig. 1c. As a key parameter for force sensing of the RSHTS, the sensitivity ($S$) is defined as $\Delta Q/\Delta F$, where $\Delta Q$ is the change in output charge of PVDF, and $\Delta F$ is the change in applied force. In the experiment, a sinusoidal force excitation is used, and amplitudes of output charge and force are adopted as $\Delta Q$ and $\Delta F$ to calculate the values of $S$ at different frequencies. We characterized the RSHTS assembled in a vibration test system by a sinusoidal force at an amplitude of 0.009–4.3 N under 5–600 Hz (Supplementary Fig. 3). First, we investigated the relationship between the applied force and the sensor output charge under different frequencies (Supplementary Fig. 4). The results show that the highly sensitive sensor has good linearity under 4.3 N in a wide frequency range of 5–600 Hz with a minimum force detection of 0.009 N, and exhibits a good frequency response (Supplementary Fig. 5). The sensitivity of RSHTS is in a range of 135.5–346.5 pC N$^{-1}$ for the stimuli at a frequency range of 5–100 Hz, which is ~20–52 times, 45–116 times those reported in the literatures 33 and 37. For higher frequency (200–600 Hz), although the sensitivity of RSHTS decreases as the sensor is a flexible device with the natural viscoelasticity, yet it keeps good linearity even at low threshold of 0.009 N. As summarized in Fig. 1e and Supplementary Table 1, our RSHTS shows a super-high sensitivity and ultrabroad bandwidth of frequency, outperforming existing piezoelectric tactile sensors reported in the literatures[26-38]. Importantly, the new working mode ($d_{31}$) of the proposed RSHTS breaks the theoretical sensitivity of traditional working mode ($d_{33}$) of existing flexible tactile sensors. In this work, the maximum sensitivity of RSHTS is 346.5 pC N$^{-1}$ (@ 30 Hz), which is ~17 times compared with theoretical value ($d_{33} = 21$ pC N$^{-1}$) (detailed proof will be shown in the following part). It is noteworthy that the same magnified effect can be achieved if the RSHTS uses other flexible piezoelectric materials.

### Working mechanism of the RSHTS

The most common design among the piezoelectric flexible tactile sensors involves bonding a PVDF film with patterned electrodes to a rigid substrate. Upon a touch loading, the sensor mainly works in $d_{33}$ mode to detect the external stimuli (Supplementary Fig. 2)[3,14,27,29,41-43]. The sensitivity $S_0$ of this sensor is mostly determined by the piezoelectric coefficient of the material (due to the in-plane normal stress $\sigma_{11}$ is neglected compared to the out-plane normal stress $\sigma_{33}$), and can be

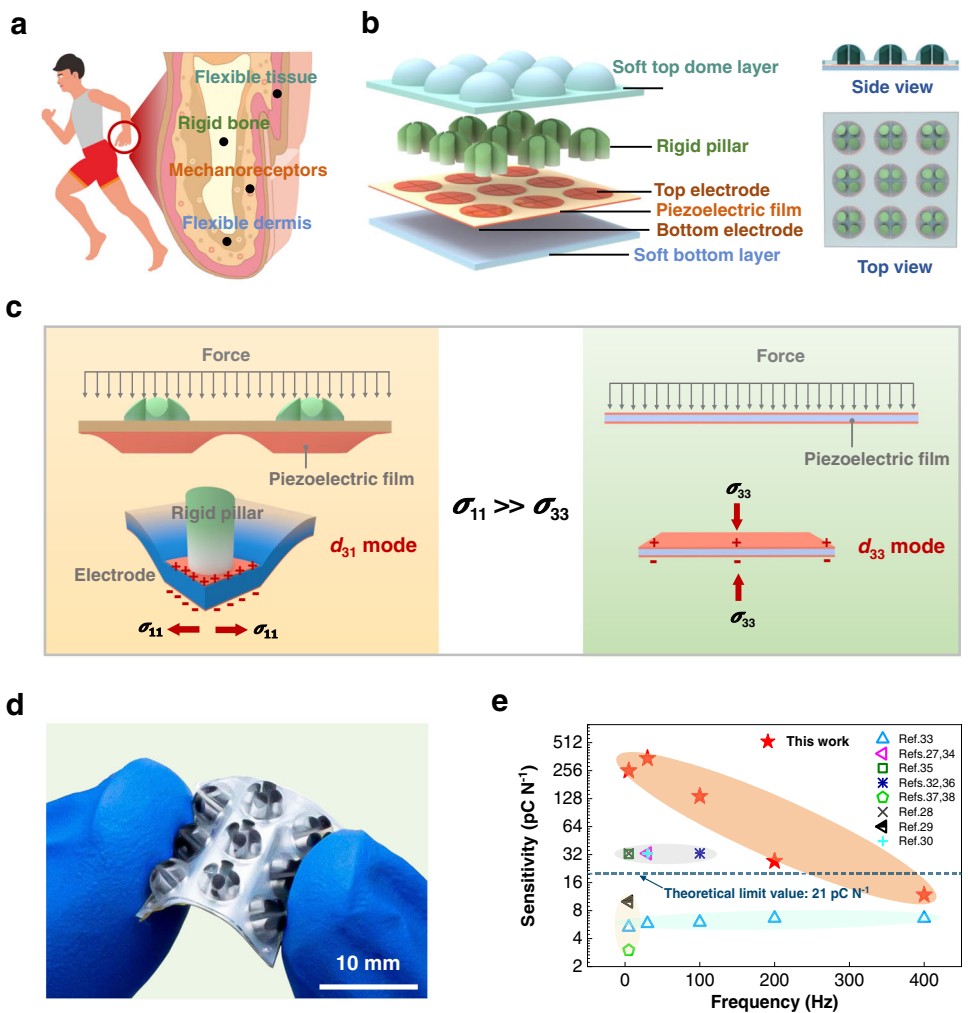

**Fig. 1 | Concept, structure and sensing performance of the RSHTS. a** Illustration of human body and anatomical structure of finger. **b** Finger-inspired rigid-soft hybrid piezoelectric tactile sensor array. **c** Differences of piezoelectric film deformation and working mode between our tactile sensor and existing piezoelectric tactile sensors. The blue color indicates the stress distribution of piezoelectric film. **d** Photograph of a 3 × 3 fabricated RSHTS array. **e** Comparison of the sensitivity of our tactile sensor with existing piezoelectric tactile sensors.

expressed as

$$S_0 = \frac{\triangle Q}{\triangle F} \approx \frac{d_{33}\sigma_{33}^0 A}{F} = \frac{d_{33}kF}{F} = d_{33}k \leq d_{33}, (k \leq 1) \tag{1}$$

Different from the common piezoelectric tactile sensors, our RSHTS using a rigid-soft hybrid force-transmission-layer in combination with a soft bottom substrate not only improves the transmission efficiency of the external force $F$, but also magnifies the effect of in-plane normal stress ($\sigma_{11}$) making the sensory layer work in $d_{31}$ mode, and its sensitivity $S$ can be expressed as

$$\frac{S}{S_0} = \frac{(d_{31} + d_{32})\sigma_{11} + d_{33}\sigma_{33}}{d_{33}\sigma_{33}^0} \tag{2}$$

where $A$ is the area of the electrode, $k$ is the transmission coefficient of the external force $F$ from the outer top layer to inner sensory layer, and $d_{31}$, $d_{32}$, $d_{33}$ are the piezoelectric strain coefficients equal to 17 pC N$^{-1}$, 6 pC N$^{-1}$, −21 pC N$^{-1}$ for the PVDF film used in this work, respectively. $\sigma_{11}$ and $\sigma_{33}$ is the in-plane normal stress and out-plane normal stress of the piezoelectric sensory layer (Supplementary Fig. 2). Detailed derivation can be seen in Supplementary Information.

From Eq. (1), we can see that the theoretical limit sensitivity of conventional piezoelectric tactile sensor is $d_{33}$. However, from Eq. (2),

the sensitivity of the sensor using a rigid-soft hybrid design depends not only on the piezoelectric coefficients, but also on the in-plane normal stress $\sigma_{11}$. For our RSHTS, $\sigma_{11}$ is significantly magnified, which is the essential reason why our sensor sensitivity can be significantly improved. This magnified effect can be applied for the flexible tactile sensor using other piezoelectric materials.

Here, we validate the $\sigma_{11}$-induced super-high sensitivity of RSHTS using finite element analysis (FEA). Due to the symmetry of RSHTS, taking one pillar and one piezoelectric capacitor as the FE model of RSHTS (Fig. 2a(iii)). To prove the function of such a rigid-soft hybrid design, the FE models of two control sensors without rigid pillar (one using a rigid bottom substrate as Control I (Fig. 2a(i)), and the other using a soft bottom substrate as Control II (Fig. 2a(ii))) are also established. Detailed simulation parameters are shown in Supplementary Table 2.

Under a normal force, $\sigma_{11}$ and $\sigma_{33}$ (averaged in-plane and out-plane normal stresses on the PVDF film in the pillar zone of three models) can be obtained from the stress analysis (Supplementary Fig. 6). As shown in the results of static analysis presented in Fig. 2b and Supplementary Fig. 6, $\sigma_{11}$ of RSHTS is greatly magnified using the rigid pillar with soft substrate, which can be improved from 0.26 kPa to 10.0 kPa (Fig. 2b), and correspondingly $|\sigma_{11}/\sigma_{33}|$ is enhanced from 0.5 to 5 (Supplementary Fig. 6). As a result, the working mode of the sensor is changed from $d_{33}$ ($\sigma_{33}$ dominantly) to $d_{31}$ ($\sigma_{11}$ dominantly), meanwhile, $\sigma_{33}$ of

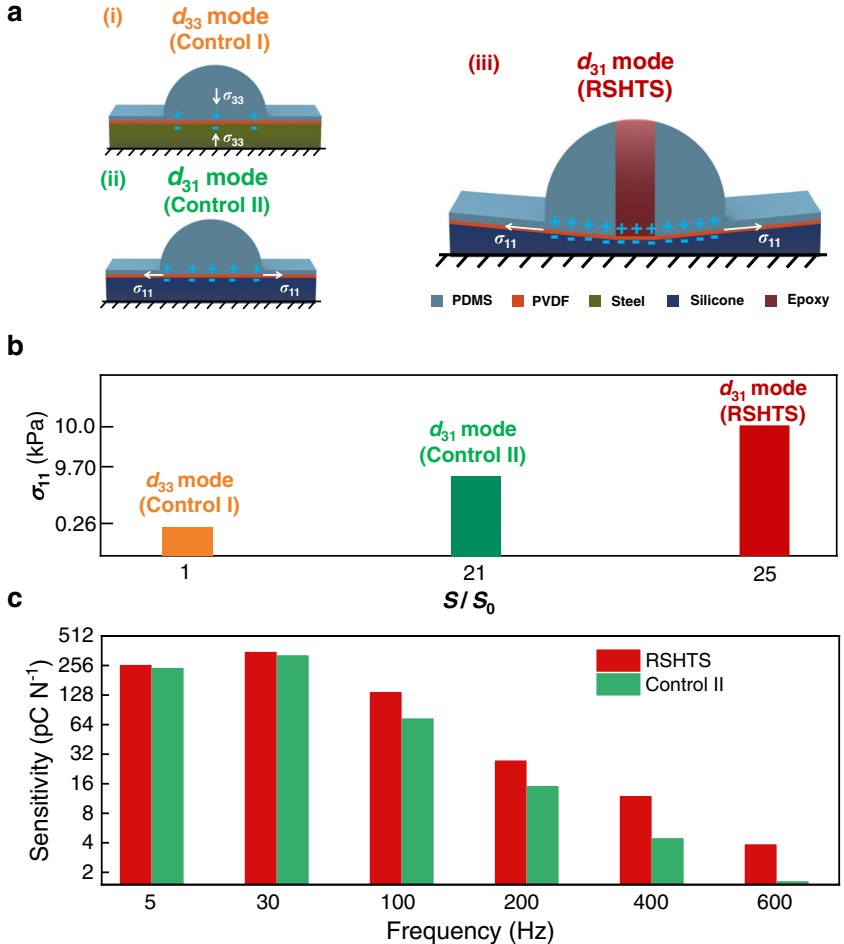

**Fig. 2 | Structural design consideration and numerical simulation for stress analysis of the RSHTS. a** Simulation models of control sensors with rigid (i, Control I) and soft substrates (ii, Control II), and our RSHTS (iii). **b** Numerical simulation results of control sensors with rigid and soft substrates, and our RSHTS. **c** Comparison of the sensitivity between the RSHTS (iii) and the control sensor without pillar (ii) in a frequency range of 5–600 Hz. The frequency response curves can be seen in Supplementary Fig. 8.

RSHTS is also improved to four times compared with that of the control sensor without the rigid pillar (Control I) (Supplementary Fig. 6). Thus, the sensitivity of RSHTS could be greatly magnified ($S/S_0 = 25$) compared with that of the traditional $d_{33}$ mode piezo-electric tactile sensor based on Eqs. (1) and (2). Moreover, we compare the sensitivity between the RSHTS with rigid pillar and the control sensor without rigid pillar (Control II) under dynamic excitation (5–600 Hz). From Fig. 2c and Supplementary Fig. 8, the experimental results prove that using the rigid pillar can effectively enhance the sensor's sensitivity for high-frequency dynamic excitation (≥100 Hz), e.g., about three times of the control sensor without the rigid pillar (Control II) at 400 Hz, indicating that the rigid pillar is the key design for sensitivity enhancement under dynamic excitation.

Thus, with the design of the rigid pillar in combination with the soft bottom substrate, the RHSTS can work in a new working mode ($d_{31}$), and its sensitivity could be greatly enhanced compared with the traditional $d_{33}$ mode piezoelectric tactile sensor under high-frequency dynamic excitation. It should be pointed out that the experimental sensitivity of our sensor is based on a preliminary design, and its sensitivity can be further improved by optimizing the parameters of the pillar and the bottom substrate as shown in simulation results (Supplementary Fig. 9).

## Performance of the super-sensitive RSHTS

In human fingers, there are more fast mechanoreceptors (FA-I, II) than slow mechanoreceptors (SA-I, II), which benefits the detection of dynamic mechanical signals in range of 5–400 Hz[5]. Besides the high sensitive dynamic force detection, the force direction recognition, stability and repeatability of flexible tactile sensor are also key technical indicators for practical applications[1,39]. Hence, the performance of the RSHTS is systematically investigated.

The RSHTS can recognize multiple force directions using one tactile unit (Fig. 3a). When a normal force in −Z axis is applied, the dome with rigid pillars is compressed and the four piezoelectric capacitors are subjected to the same compressive stress. Due to the piezoelectric effect of PVDF film, charges with the same polarity (negative) are generated on the four piezoelectric capacitors. When a shear force, such as in X axis is applied, the dome tilts and generates a torque at the fixed end. The two piezoelectric capacitors on the left side are subjected to compressive stress, whereas other two on the right side are subjected to tensile stress. As a result, the developed charges on the left electrodes (negative) and right electrodes (positive) are of opposite polarity. Similar analysis can be used to recognize the direction of shear force in −X and ± Y axes. From Fig. 3a and Supplementary Fig. 10, taking one tactile unit as an example, the RSHTS can correctly determine the direction of the force in ± X, ±Y, and ± Z axes by monitoring the polarities of the generated charges on the four piezoelectric capacitors ($C_1$, $C_2$, $C_3$, and $C_4$). By contrast, it is difficult to use the control sensor without a rigid pillar for determining the force direction due to almost no charge generated by the four piezoelectric capacitors for the same excitation, compared with RSHTS (Supplementary Fig. 11). This also

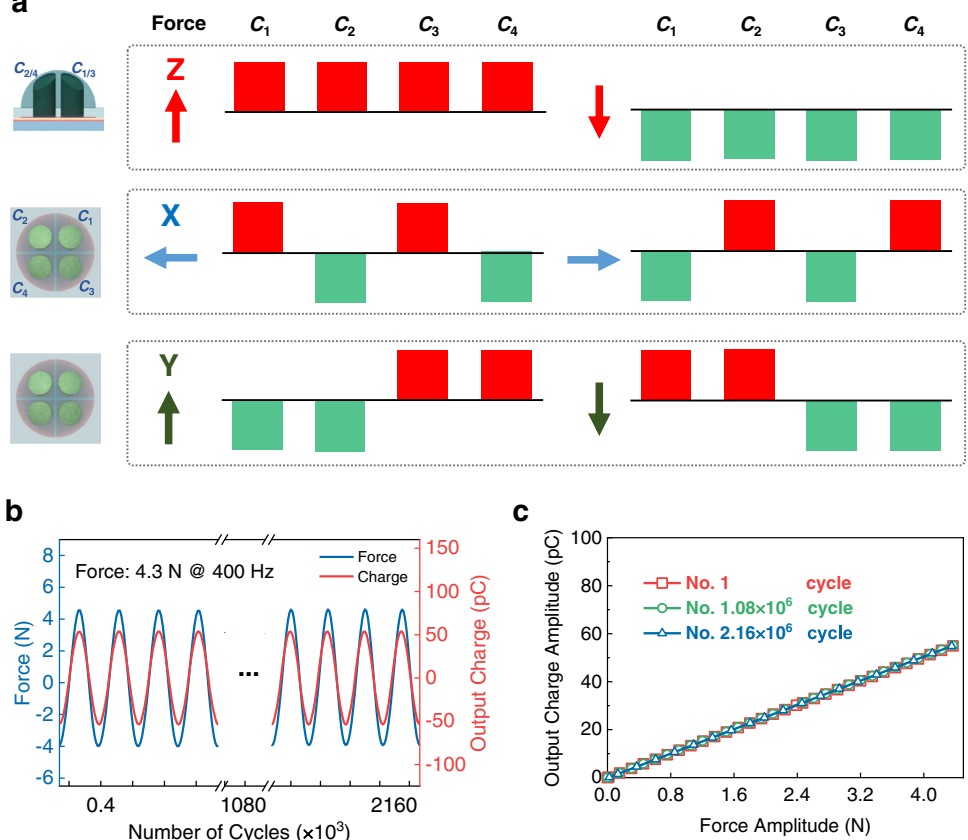

**Fig. 3 | Force direction detection, cycling stability, and repeat performance of the RSHTS. a** Force direction detection by one sensory unit. **b** Signal stability over $2.16 \times 10^6$ cycles under a sinusoidal force of 4.3 N and 400 Hz. **c** Output charge of RSHTS as a function of applied normal force measured at 400 Hz is repeated after 1, $1.08 \times 10^6$ and $2.16 \times 10^6$ cycles, and the sensitivity is correspondingly presented as 12.58, 12.64, and 12.64 pC N$^{-1}$.

proves the effectiveness of such a rigid-soft hybrid design for sensor sensitivity improvement.

Moreover, the RSHTS exhibits good durability during the cycling test: the output charge remains highly stable over $2.16 \times 10^6$ cycles excited by a repeated sinusoidal loading of 4.3 N @ 400 Hz (Fig. 3b), which is also confirmed in other cases with different frequencies under normal and shear forces (Supplementary Figs. 12–14). Furthermore, the sensitivity change is below 0.5% for $2.16 \times 10^6$ cycles at 400 Hz (Fig. 3c), and 1.0% for $5 \times 10^4$ cycles at 5 Hz (Supplementary Fig. 15), showing the outstanding repeatability of RSHTS. Hysteresis is considered as one of parameters to ensure practical applications of the flexible tactile sensor[44]. Our results suggest that the proposed RSHTS presents a low hysteresis error (≤5%) in a range of 5−600 Hz (Supplementary Figs. 16, 17). Finally, a RSHTS array is demonstrated for pressure mapping (Supplementary Fig. 18). The array is $1.6 \times 1.6$ cm$^2$ in an area consisting of 36 piezoelectric capacitors, and result shows that RSHTS array is capable of resolving pressure at a millimeter scale.

## RSHTS-based robotic hand and its applications in tactile perception

To demonstrate the potential applications of RSHTS, the RSHTS array is attached to a human finger and robotic hand to detect high-frequency signals that simulate the human-like tactile system.

The RSHTS responses fast and is able to resolve different frequency signals. A $3 \times 3$ RSHTS array is mounted on the human finger for simulating the tactile system of robots. Here, human hand holds three tuning forks with different natural frequencies, and the output charge of RSHTS after multiple shock excitations is recorded in Supplementary Video 1. As shown in Fig. 4a and Supplementary Fig. 19, the experimental results show that RSHTS can recognize vibration

frequency (129, 257, and 513 Hz) and change process of the external dynamic signal. It is expected that using RSHTS to recognize different vibration signals through detecting frequency, could establish a sensitive tactile system for robots and the handicapped or aged people with weak tactile sensing to feel the information from the environment, such as phone vibration.

The RSHTS array can detect force direction due to the four-electrode structure in combination with a dome-shape layer. This is important for a transport robot system since recognizing the collision force helps robot to adjust its next motion effectively when encountering obstacles. Here, we use the RSHTS-based robotic hand to hold a mass block. The block was knocked by a hammer from different directions (Supplementary Fig. 20, Video 2) to simulate the robot's moving object and encountering a collision. Taking one sensory unit as an example, the real-time output of four piezoelectric capacitors is recorded shown in Fig. 4b. The output charge polarity changes with knocking direction. If the hammer knocks down the mass block, a downward torque is generated, as a result, $C_1$ and $C_2$ are subjected to tensile stress and generate positive charges, whereas $C_3$ and $C_4$ are subjected to compressive stress and generate negative charges. Thus, other collision directions can also be recognized by analyzing the output charge polarity. Here, the positive and negative peaks of $C_1$, $C_2$, $C_3$, and $C_4$ are labeled to the binary values of 1 and 0, respectively. So, the decimal value of $C_1C_2C_3C_4$ can be used to present the knock directions shown in the table (Fig. 4b). The charge polarity characteristic of $C_1$, $C_2$, $C_3$, and $C_4$ to force direction is also successfully used to identify slipping direction during grasping process by a robot arm with RSHTS embedded (Supplementary Fig. 21).

As shown in Fig. 4c, the RSHTS array is attached to the robotic hand for monitoring the process of robot grasping a bottle and

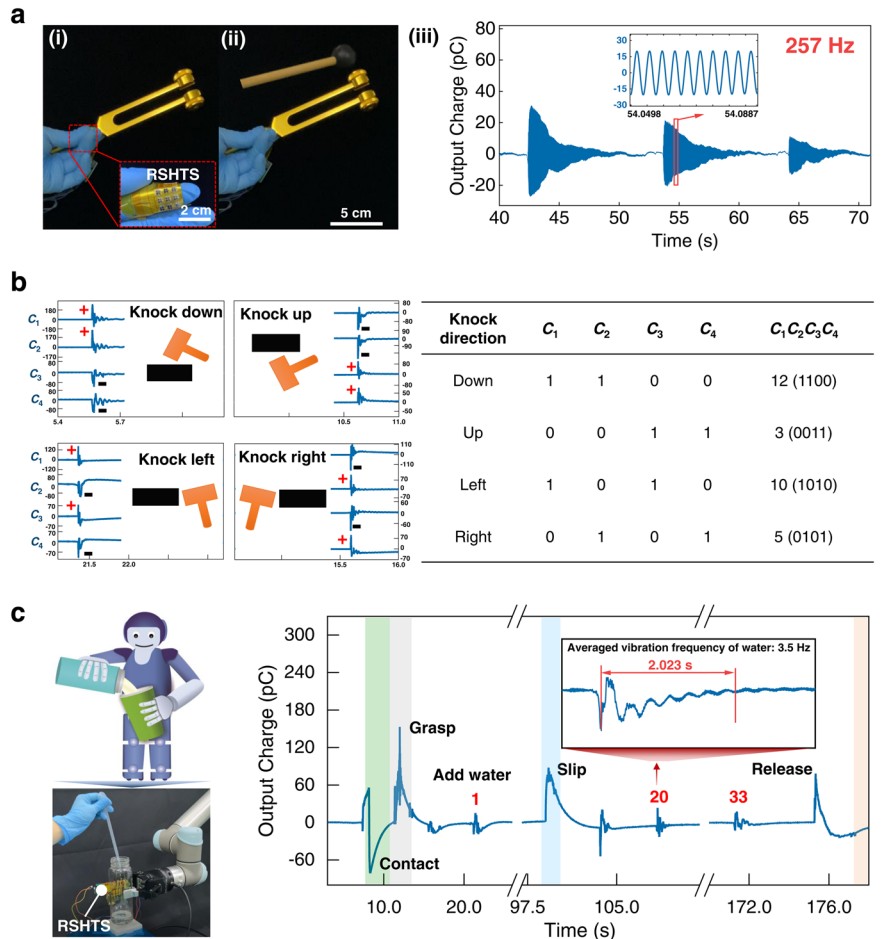

**Fig. 4 | High-frequency stimuli detection of the RSHTS and RSHTS-based robotic hand to detect impact force, and simulate robot pouring water. a** High-frequency waveforms generated by tuning forks are recorded by the RSHTS attached to the human finger. Other high-frequency waveforms are shown in Supplementary Fig. 19. **b** The real-time output charge of four piezoelectric capacitors, from which the knock directions can be determined. Here, the positive and negative peaks of $C_1$, $C_2$, $C_3$, and $C_4$ are labeled to binary values of 1 and 0, respectively. Thus, the decimal value of $C_1C_2C_3C_4$ can be used to present the knock directions shown in the table. **c** A RSHTS-based robotic hand grasps and holds a plastic bottle, into which water drops are added to simulate the scenario of a robot pouring water. Real-time output charge of one piezoelectric capacitor (others shown in Supplementary Fig. 22), from which the motions of the robotic hand catching bottle, grasping bottle, receiving water, and releasing bottle can be identified. The subgraph shows the details of water vibration at a frequency of 3.5 Hz caused by falling droplets.

pouring water, which is easy for a healthy human, but hard for the robot or handicapped. We simulate the water-pouring process: a bottle is initially placed on a platform, then the robotic hand contacts and grasps the bottle, and water are added drop by drop, finally, the bottle is released (Supplementary Video 3). From Fig. 4c, it can be seen that the whole dynamic process is clearly recorded by the RSHTS. All these data indicates that the RSHTS is able to detect external stimuli in different application scenarios, endowing robot with a human-like tactile system for high dexterity operation.

## Discussion

Inspired by the structures of finger of animals and humans—skeleton embedded in muscle, a super-high sensitive tactile sensor using a rigid-soft hybrid force-transmission-layer in combination with a soft bottom substrate, which overcomes the dynamic sensitivity limit for conventional piezoelectric flexible tactile sensors. Instead of working in $d_{33}$ mode for conventional sensors, our RSHTS realizes sensing in $d_{31}$ mode that achieves a much higher sensitivity. More interestingly, our RSHTS also possesses flexibility comparable to that of the traditional flexible sensors.

The working mechanism of RSHTS is clarified. The results show that the rigid-soft hybrid force-transmission-layer in combination with a soft bottom substrate of the sensor, on the one hand, improves the transmission efficiency of the force from the outer to inner sensory layer, which is four times compared with that of the conventional $d_{33}$ mode sensor. On the other hand, a significantly enhanced in-plane normal stress triggers $d_{31}$ mode of the piezoelectric sensory layer, resulting in a super-high sensitivity. The sensor exhibits a high sensitivity of 346.5 pC N$^{-1}$ at 30 Hz, which is ~17 times the theoretical limit sensitivity in the $d_{33}$ mode (21 pC N$^{-1}$). Moreover, the reliable RSHTS has a wide bandwidth of 5–600 Hz and a linear force detection range of 0.009–4.3 N.

Combining the design of the dome in the top layer and patterned electrodes in the sensory layer, the RSHTS can detect the multiple force directions by analyzing the output of four piezoelectric capacitors. The RSHTS array using one sensory unit can distinguish the direction of applied force in ±X, ±Y, and ±Z axes, while it is difficult to determine the force direction for the control sensor without rigid pillar due to the limited sensor sensitivity. Furthermore, the RSHTS can clearly recognize high-frequency vibration, demonstrating the advantage for dynamic force detection. A RSHTS-based robotic hand is used to detect the impact force and monitor the process of pouring water, which shows the great potential of RSHTS to help robots achieve high dexterity operation. The proposed sensor is expected to be applied for

wearable electronics to enable long-term monitoring of external stimuli for establishing a human-like tactile system of the robot, and recovering tactile sensing capability of the disabled or elder people, and so forth.

## Methods

### Fabrication of rigid-soft hybrid top layer

The rigid-soft hybrid substrate with rigid pillars (four pillars in one sensory unit, one pillar with 0.6 mm in radius, its arc upper surface with 1.9 mm in radius, 1.5 mm in pitch) and domes (2 mm in radius, 5 mm in pitch) is prepared in seven steps (Supplementary Fig. 23). One template with $3 \times 3$ pit (named as pit mold), and the other one with pillars (named as pillar mold) are fabricated by a 3D printer with a printing resolution of 10 μm (nanoArch® S140, BMF Precision Tech Inc., China) (Supplementary Fig. 23a). A mixture of Sylgard 184 base and curing agent in a weight ratio of 10:1 (Dow Corning Co., Ltd) is cast onto the pit mold (Supplementary Fig. 23b). Before the PDMS solidified, the pillar mold is put into pit mold, then cured at 60 °C for 24 h after 30 min still standing (Supplementary Fig. 23c, d). The pillar mold is removed first, thus, holes appear in each sensory unit (Supplementary Fig. 23e). Then a mixture of Epoxy A and B in a weight ratio of 3:1 is cast onto the PDMS holes, and cured at 25 °C for 24 h (Supplementary Fig. 23f). The cured $3 \times 3$ dome-shaped PDMS with rigid pillars (detailed parameters are shown in Supplementary Fig. 23h) is slowly peeled off from the pit mold, and is cleaned by alcohol to remove excess materials (Supplementary Fig. 23g).

### Fabrication of patterned PVDF

Supplementary Fig. 24 shows the fabrication process for the patterned PVDF, as follows: (1) A 30 μm-thick PVDF film ($20 \times 27$ mm$^2$) (Jinzhou Kexin Electronic Materials Co., Ltd, China) with 220 nm-thick Al layer is spread on a 4-inch silicon wafer and fixed by tape (Supplementary Fig. 24a). (2) Next, a 2.5 μm-thick positive photoresist layer (RZJ-304-50, Suzhou Ruihong, China) is spin-coated onto the top surface of Al at 5000 rpm and baked for 60 min at 60 °C in an oven (DZF-6020, Jinghong Shanghai, China) (Supplementary Fig. 24b). (3) The photoresist-coated PVDF film is exposed to lithography device (MA6, SUSS, Germany) with 600 mW cm$^{-2}$, and developed for 30 s in developer (AZ326MIF, Microchem, USA). (4) The developed film is rinsed, dried, and baked for 5 min at 60 °C in an oven. The patterned photoresist on the Al surface is used as a mask for the next Al wet-etching (Supplementary Fig. 24c). (5) The PVDF-silicon wafer is immersed into the Al etchants (TechiEtch Al80, MicroChem, USA) for 2 min. Then, a patterned Al layer is shaped on the PVDF film (Supplementary Fig. 24d). (6) Last, after removing the photoresist with acetone and stripping the tape, the PVDF film with patterned Al electrodes is fabricated (Supplementary Fig. 24e).

### Fabrication of RSHTS array

The liquid-state PDMS was spin-coated onto the top surface of sensory layer to bond the top rigid-soft hybrid layer, also spin-coated onto the bottom surface of sensory layer to bond the soft bottom layer. Thus, a $3 \times 3$ RSHTS array with an area of $16 \times 16$ mm$^2$ is fabricated.

### Measurements

As shown in Supplementary Fig. 3, the RSHTS array is fixed on a three-axis positioner to realize the alignment between the axis of loading bar and the dome center of a sensory unit. The loading bar is attached to a six-axis force sensor (Nano43, ATI Industrial Automation, Apex, NC, USA) which is installed on the shaker and used to measure the external force loaded on the sensor. The external force is supplied by a shaker (GW V20, DP, USA) of which frequency and amplitude are controlled by a signal generator (33220 A, Agilent Technologies, Palo Alto, CA, USA) and a power amplifier (PA100E, DP, USA). Charges generated by

the RSHTS array are recorded by a dynamic measurement system (DH5983, Donghua Testing Technology Co., Ltd., China). To achieve a pure sinusoidal force excitation, before all the dynamic tests, a preload force of 5.0 N and 4.0 N is applied on RSHTS and control sensor (Control II), respectively. The measurement charge and force data generated in this study are provided in the Supplementary Information.

### Finite element simulation

FEA is conducted using Abaqus/Standard 2019. We numerically simulate the force-controlled compression test of RSHTS and control sensors using the axisymmetric model. The PDMS, silicone, pillar and PVDF are modeled as linear elastic with Young's modulus $E_{PDMS} = 2.0$ MPa, $E_{Silicone} = 1.0$ MPa, $E_{Pillar} = 4.1$ GPa, $E_{PVDF} = 3.0$ GPa, and Poisson's ratio $v_{PDMS} = 0.46$, $v_{Silicone} = 0.4$, $v_{Pillar} = 0.32$, $v_{PVDF} = 0.35$. A compressive force of 5.0 mN is applied on the top layer. All interfacial contacts are assumed to be tied together. The stress data generated in simulation are provided in the Supplementary Information.

## Data availability

The data that support the findings of this study are provided in the Supplementary Information.

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

## Acknowledgements

We acknowledge support from the National Natural Science Foundation of China (no. 51922092, no. 52005423, no. U21A20136), Fundamental Research Funds for the Central Universities (no. 20720200068), China Postdoctoral Science Foundation (no. 2020M671946), and Health@InnoHK (Hong Kong Centre for Cerebro-cardiovascular Health Engineering (COCHE)). The authors thank professor Zhong-Qun Tian, professor Wenjing Hong, Dr. Xuyang Chu, Dr. Yunsong Lian, Dr. Shaogan Ye, Dr. Yanming Xia, Huiran Zhang, Maoyu Lin, Jigang Ge, Hu Xia, Chun Yang, Tingting Lian, Jincheng Wang, Da Geng, Renpeng Wang, Xinying Zhu, Xiaodong Wu, and Linjing Wu for the help during the research.

## Author contributions

W.Z., L.F.Q., and Z.K.W. supervised the research. J.H.Z., W.Z., and L.F.Q. conceived the ideal. J.H.Z. designed and performed the experiments. J.H.Z., H.M.Y., S.Y.C., and Y.X. performed the theoretical analysis. J.H.Z. and S.L.M. fabricated the device. J.H.Z., J.Y.M., R.C., T.L., and W.S.L. analyzed the data. J.H.Z., L.F.Q., W.Z., S.Y.C., H.M.Y., and Z.K.W. drafted the manuscript, and all authors contributed to writing and revising the manuscript.

## Competing interests

The authors declare no competing interests.
