## [Peer Review File · Nature Communications]

REVIEWER COMMENTS

Reviewer #1 (Remarks to the Author):

Title: Finger-Inspired Rigid-Flexible Coupled Tactile Sensor with Superior Sensitivity at High Frequency

The paper examines a novel rigid-flexible coupled tactile sensor (RFCTS) with both high sensitivity and high frequency in measurement. The contact sensor uses four rigid pillars in a flexible substrate with piezoelectrical sensors as the force transmission layer. The sensitivity is greatly improved by utilizing in-plane stress to magnify the signal given from the received force. The data showed the sensitivity in d31 direction is convincing to support the proposed working mechanism. However, the sensor has major shortcomings and requires further supporting data to strengthen the claim of the high-frequency capability of the tactile sensor. The quality of the paper can be improved if the author addressed the following major and minor comments:

Major Comments:

1. The rigid-flexible coupled design of the tactile sensor does not look novel. Other tactile sensors, for example the paper by Vogt et al. [1], might not utilize the polyvinylidene fluoride (PVDF) layer as sensing media for improving the sensitivity of the tactile sensor, but it might still utilize in-plane stress through the presence of a rigid skeleton to enhance the tactile sensor's sensitivity using other types of sensing media. Further clarification is needed on the novelty of the tactile sensor. The author needs to clarify the main differences between their tactile sensor and other soft tactile sensors (such as the sensor in [1]) that utilize rigid skeleton to improve force transmission.

2. The authors claim that "This also proves the rigid-flexible coupled design can efficiently improve the sensitivity of the sensor" on Page 7. The Purely Flexible Tactile Sensor (PFTS) may rely on the design of the receptor shape to transmit the force to the force-sensing media. It is not accurate to suggest the PFTS is not efficient in its force sensing capability by comparing it to a similar receptor shape that yields good results with the (rigid-flexible coupled tactile sensor) RFCTS. Other receptor shapes should be studied to further prove the author's claim.

3. "the RFCTS remains highly stable over 2.16×10^6 cycles excited by a repeated sinusoidal loading of 4.3 N @ 400 Hz (Fig. 3b), which was also confirmed by the cases of 0.9 N @ 400 Hz and 4.2 N @ 5 Hz" What is the reasoning for the selected force and frequency? What direction of force is used for

this experiment? Does this result differ with different force directions and combinations of compression and shear force? Additional data is needed to support the author's claim on the stability of the tactile sensor's response.

4. "Our results suggest that the proposed RFCTS presents a low hysteresis error of 4%" What frequency is this hysteresis acquired on? Due to the viscoelastic effect of the soft PDMS and silicone layer of the RFCTS, changes in the strain rate due to different frequencies should cause variability in the hysteresis error. Instead of presenting it as a table (Supplementary Table 3), the author could present the hysteresis data in form of hysteresis loops Input force vs RFCTS output charge for different frequencies.

5. On page 9, "Thus, other collision directions can also be recognized by analyzing the output charge polarity. Here, the positive and negative peaks of C1, C2, C3 and C4 are labeled to binary value of 1, 0, respectively. So, the decimal value of C1C2C3C4 can be used to present the knock directions shown in the table (Fig. 4b)." The data shown in Figure 4b looks incomplete. Can the author decouple the force magnitude and direction using the four inputs (C1, C2, C3, and C4)? This finding is crucial as the viscoelastic effect due to the high-frequency operation of the tactile sensor might affect the delayed response of the four signals.

Minor Comments:

1. "To mimic the diverse mechanical sensory function of finger, commonly used approaches for converting tactile information into electrical signals include piezoresistive, capacitive, piezoelectric, and triboelectric types." This statement suggests the aforementioned sensors aimed to mimic the mechanical sensory function of the finger, although this might not be the motivation for the design of these sensors. The author may consider rewriting this sentence.

2. "Their form often takes a layered structure of two flexible substrates sandwiching a sensory layer, where one of the flexible substrates is used to contact stimuli directly and transfers force from outer to inner sensory layer" This statement is not complete as some soft sensor uses soft and flexible substrate to allow the sensor to conform to curved surfaces [2] since some application involves interacting with a human, not just to transmit the stimuli to the sensory layer.

3. What is the supplier of the PVDF layer is used for the fabrication of the tactile sensor? If this layer is fabricated by the authors, please explain the fabrication process.

References

[1] Vogt DM, Park Y-L, Wood RJ. Design and Characterization of a Soft Multi-Axis Force Sensor Using Embedded Microfluidic Channels. IEEE Sens J 2013; 13: 4056–4064.

[2] Dahiya RS, Mittendorfer P, Valle M, et al. Directions toward effective utilization of tactile skin: A review. IEEE Sens J 2013; 13: 4121–4138.

Reviewer #2 (Remarks to the Author):

This manuscript is on the investigation of a novel flexible tactile sensor working at high frequency for wearable electronics and intelligent robotics applications. The authors got their inspiration from our human finger's structures, rigid skeleton embedded inside flexible muscle, to design a unique rigid-flexible coupled tactile sensor structure to achieve superior sensitivity. The authors also carried out extensive experiments and the results show that 17 times sensitivity enhancement can be achieved compared with the theoretical limit value in d33 mode. The physics behind such high performance is attributed to 1) enhanced transmission of high frequency dynamics force as well as 2) magnified effect of piezo-electric sensory layer in d31 mode. This piece of research work is very interesting and impressive with highly potential applications. Such design to achieve super sensor performance can be a very good reference for future sensor designs and researches.

I would like to recommend the Minor Revision of the manuscript before it can be accepted for publication. The authors need to address the following issue properly:

1. There are some English grammar errors inside the manuscript. The authors should correct them carefully. I scanned the reviewed copy and upload the files for their reference.
2. Could such structure' performance be further improved by using other flexible materials as well as optimized rigid array structures?
3. Since the rigid structures are embedded inside flexible layers, does it form cracks after many runs of testing, typically the interfaces between flexible layer and rigid surface are the places where the mechanical stress/cracks can be formed, and such cracks grow with many runs of operating. It finally affects the structure' reliability.
4. Supplementary Fig. 7a, Z axis direction needs to be corrected.

Response Letter

We thank the reviewers for their helpful and constructive comments on our manuscript. Below, we provide our responses to each reviewer's comments. The original comments are in italics, and our responses are in normal text and blue colour. The revisions in the revised manuscript and Supporting Information are highlighted in yellow background.

Reviewer #1:

The paper examines a novel rigid-flexible coupled tactile sensor (RFCTS) with both high sensitivity and high frequency in measurement. The contact sensor uses four rigid pillars in a flexible substrate with piezoelectrical sensors as the force transmission layer. The sensitivity is greatly improved by utilizing in-plane stress to magnify the signal given from the received force. The data showed the sensitivity in d_{31} direction is convincing to support the proposed working mechanism. However, the sensor has major shortcomings and requires further supporting data to strengthen the claim of the high-frequency capability of the tactile sensor. The quality of the paper can be improved if the author addressed the following major and minor comments:

Response: We sincerely thank the reviewer for his/her positive comment on our manuscript. The point-by-point response is as follows.

Major Comments:

1. The rigid-flexible coupled design of the tactile sensor does not look novel. Other tactile sensors, for example the paper by Vogt et al. [1], might not utilize the polyvinylidene fluoride (PVDF) layer as sensing media for improving the sensitivity of the tactile sensor, but it might still utilize in-plane stress through the presence of a rigid skeleton to enhance the tactile sensor's sensitivity using other types of sensing media. Further clarification is needed on the novelty of the tactile sensor. The author needs to clarify the main differences between their tactile sensor and other soft tactile sensors (such as the sensor in [1]) that utilize rigid skeleton to improve force transmission.

Response: We thank the reviewer for sharing this important work by Vogt et al. [1]. Actually, the sensor reported in our work is quite different from that proposed by Vogt et al. [1], especially in terms of force transformation. The piezoresistive sensor reported in [1] applies liquid metal as the sensory media, which is incompressible and highly mobile. Under external pressure stimulus, the liquid metal encapsulated in a microscopic channel tend to flow into regions where the pressure is lower, leading to the change of the channel morphology and, therefore, the variation of electric resistance (see Figs. R1a, b). The applied pressure is sensed by monitoring the variation of the liquid metal's electric resistance.

However, in our design, we use PVDF film, a solid piezoelectric polymer sandwiched in elastomers, as the sensory media. Upon external pressure in Direction 3, the elastomers tend to extend in Direction 1 due to their large Poisson's ratios (see Fig. R1c). As a result of the mechanical constraint by the stiffer PVDF layer to such extension, considerable shear stress is developed on the interfaces between the elastomer and PVDF film, which then results in significant in-plane normal stress σ_{11} in the PVDF film due to its ultra-small thickness. As a result, the PVDF film works in the d_{31} mode instead of the traditional d_{33} mode, and the sensor exhibits extremely high sensitivity in a new working mode. The experimental results show that the sensitivity of RSHTS (346.5 pC N^{-1} @ 30 Hz) can be about 17 times the theoretical sensitivity of d_{33} mode

(21 pC N⁻¹), about 52 times and 116 times those reported in existing literatures 33 (6.62 pC N⁻¹) and 37 (3.0 pC N⁻¹).

According to the reviewer’s suggestion, revision has been made to the introduction and results of the manuscript (see Pages 2–3 and 5–7), and related reports have been cited in references 6 and 9.

6. Vogt, D. M., Park, Y. L. & Wood, R. J. Design and characterization of a soft multi-axis force sensor using embedded microfluidic channels. *IEEE Sens. J.* **13**, 4056–4064 (2013).
9. Park, Y. L., Majidi, C., Kramer, R., Brard, P. & Wood, R. J. Hyperelastic pressure sensing with a liquid-embedded elastomer. *J. Micromechanics Microengineering* **20**, (2010).

Fig. R1 **a** The soft multi-axis force sensor proposed in reference [1]. **b** Deformation of the sensor in reference [1] under a normal pressure. **c** Deformation of the rigid-soft hybrid tactile sensor (RSHTS) under a normal pressure.

2. The authors claim that “This also proves the rigid-flexible coupled design can efficiently improve the sensitivity of the sensor” on Page 7. The Purely Flexible Tactile Sensor (PFTS) may rely on the design of the receptor shape to transmit the force to the force-sensing media. It is not accurate to suggest the PFTS is not efficient in its force sensing capability by comparing it to a similar receptor shape that yields good results with the (rigid-flexible coupled tactile sensor) RFCTS. Other receptor shapes should be studied to further prove the author’s claim.

Response: Thank you for this valuable suggestion. We agree that the performance of the purely flexible tactile sensor (PFTS) relies on the design of the receptor shape and that better performance could be achieved by optimising the receptor shape of the upper flexible substrate (Fig. R2).

The abbreviation of PFTS in our manuscript is only used to describe the control group of our design, which does not cover other PFTSs reported previously. To clarify our point, we revised the PFTS without rigid structure into a ‘control sensor, and the sentence ‘This also proves the rigid-flexible coupled design can efficiently improve the sensitivity of the sensor’ has been rewritten as follows:

‘This also proves the effectiveness of such a rigid-soft hybrid design for sensor sensitivity improvement.’

We found that introducing a rigid structure can further enhance the sensitivity of the original PFTS through intensified force transmission. As shown in Fig. R2, in a view of static analysis, the sensitivities of the flexible tactile sensors without a rigid pillar and with three different upper receptor shapes can be enhanced by further inducing the rigid pillar. Moreover, we also compare the sensitivity between the RSHTS with rigid pillar and the control sensor without rigid pillar under dynamic excitation (5–600 Hz). As shown in the experimental results presented in Fig. 2c, using the pillar can effectively enhance the sensor’s sensitivity for high-frequency dynamic excitation (≥ 100 Hz). For example, it can be about three times (@ 400 Hz) the sensitivity of the control sensor without the pillar, indicating that the rigid pillar is the key design for sensitivity enhancement under dynamic excitation. The corresponding experimental results have been added

in the manuscript (see Pages 5–7) and as supporting information found in Supplementary Fig. 8.

Fig. R2 Numerical simulation for sensitivity S/S_0 of the RSHTS and the flexible sensor without rigid pillar (S normalised by the sensitivity S_0 of the conventional piezoelectric tactile sensor working in d_{33} mode) under different top receptors' shapes.

Fig. 2c Comparison of the sensitivity between the RSHTS (iii) and the control sensor without pillar (ii) in a frequency range of 5–600 Hz. The frequency response curves can be seen in Supplementary Fig. 8.

In addition, the sensitivity of the RSHTS can be improved by increasing the pillar height or by reducing the pillar radius, Young's modulus and the thickness of the bottom substrate (Supplementary Fig. 9). The discussion regarding the RSHTS design has been added in the 'Working mechanism of the RSHTS' part of the manuscript (see Page 6), and the corresponding results have been added as supporting information in Supplementary Fig. 9. The added part is given below:

'It should be pointed out that the experimental sensitivity of our sensor is based on a preliminary design, and its sensitivity can be further improved by optimizing the parameters of the pillar and the bottom substrate as shown in simulation results (Supplementary Fig. 9).'

Supplementary Fig. 9 | Numerical simulation for sensitivity S/S_0 of the RSHTS (S normalised by the sensitivity S_0 of the conventional piezoelectric tactile sensor working in d_{33} mode) under different parameters, including pillar (a) radius and (b) height, (c) Young's modulus and (d) thickness of the bottom substrate.

3. “the RFCTS remains highly stable over 2.16×10^6 cycles excited by a repeated sinusoidal loading of 4.3 N @ 400 Hz (Fig. 3b), which was also confirmed by the cases of 0.9 N @ 400 Hz and 4.2 N @ 5 Hz” What is the reasoning for the selected force and frequency? What direction of force is used for this experiment? Does this result differ with different force directions and combinations of compression and shear force? Additional data is needed to support the author's claim on the stability of the tactile sensor's response.

Response: Thank you for the comment. For the robot with human-like dynamic tactile sensing, the frequency range of 5–400 Hz was selected in the sensor stability test based on the frequency response range of fast tactile afferents (FA–I, II) in a human hand (reference 5 in the revised manuscript). A preload force 5.0 N for the RSHTS was applied to the sensor to achieve a pure sinusoidal force excitation for the dynamic tests. To ensure stable contact between the sensor and the shaker during the experiment, the force detection range of the sensor must be smaller than the preload force. Therefore, the force < 5 N was selected for the stability test.

To further characterise the stability of the sensor, we also tested the sensor under excitation of normal/shear force with different frequencies. As shown in Supplementary Figs. 13–14 and Tables 3–4, the test results show that our sensor has good stability. The corresponding experimental results have been added in the supporting information.

Supplementary Fig. 13 | Cycling stability of the RSHTS under a sinusoidal normal force range of 5–600 Hz.

Table 3 The change of output charge of the RSHTS under normal force with different excitation frequencies based on Supplementary Fig. 13.

Frequency (Hz)	5	30	100	200	400	600
Change of output charge (%)	3.88	2.93	0.28	0.65	1.18	0.03

* The change of output charge is calculated by the average amplitude (10 cycles) at the start and end range of the recorded data during the experiment.

Supplementary Fig. 14 | Cycling stability of the RSHTS under a sinusoidal shear force of 5–600 Hz.

Table 4 The change of output charge of the RSHTS under shear force with different excitation frequencies based on Supplementary Fig. 14.

Frequency (Hz)	5	30	100	200	400	600
Change of output charge (%)	3.52	3.50	0.84	1.87	0.87	2.98

4. “Our results suggest that the proposed RFCTS presents a low hysteresis error of 4%” What frequency is this hysteresis acquired on? Due to the viscoelastic effect of the soft PDMS and silicone layer of the RFCTS, changes in the strain rate due to different frequencies should cause variability in the hysteresis error. Instead of presenting it as a table (Supplementary Table 3), the author could present the hysteresis data in form of hysteresis loops Input force vs RFCTS output charge for different frequencies.

Response: We thank the reviewer for his/her professional suggestion. The hysteresis error of 4% of the

RSHTS in the manuscript was tested at 400 Hz. Moreover, we tested its hysteresis loops within a range of 5 - 600 Hz under both normal and shear forces. The related results have been added in the supporting information, as shown in Supplementary Figs. 16–17.

As the reviewer has mentioned, the experimental results indicate that the hysteresis error changes with the excitation frequency. This may be due to the fact that the whole device remains a flexible device, which is caused by the viscoelastic effect of the flexible material, such as the bottom silicone substrate of the RSHTS. The sentence ‘Our results suggest that the proposed RFCTS presents a low hysteresis error of 4%’ has been revised as follows: ‘Our results suggest that the proposed RSHTS presents a low hysteresis error ($\leq 5\%$) in a range of 5–600 Hz.’

Supplementary Fig. 16 | Hysteresis performance of the RSHTS under normal force with excitation frequencies ranging from 5–600 Hz.

Table 5 Hysteresis error of the RSHTS under normal force with excitation frequencies ranging from 5–600 Hz based on Supplementary Fig. 16.

Sensor	Frequency (Hz)	5	30	100	200	400	600
RSHTS	Hysteresis error (%)	0.30	1.61	0.51	0.13	0.41	0.68

Supplementary Fig. 17 | Hysteresis performance of the RSHTS under shear force (y-axis) with excitation frequencies ranging from 5–600 Hz.

Table 6 Hysteresis error of the RSHTS under shear force (y-axis) with excitation frequencies ranging from 5–600 Hz based on Supplementary Fig. 17.

Sensor	Frequency (Hz)	5	30	100	200	400	600
RSHTS	Hysteresis error (%)	1.88	5.13	0.36	1.13	0.45	1.45

5. On page 9, “Thus, other collision directions can also be recognized by analyzing the output charge polarity. Here, the positive and negative peaks of C_1 , C_2 , C_3 and C_4 are labeled to binary value of 1, 0, respectively. So, the decimal value of $C_1C_2C_3C_4$ can be used to present the knock directions shown in the table (Fig. 4b).” The data shown in Figure 4b looks incomplete. Can the author decouple the force magnitude and direction using the four inputs (C_1 , C_2 , C_3 , and C_4)? This finding is crucial as the viscoelastic effect due to the high-frequency operation of the tactile sensor might affect the delayed response of the four signals.

Response: We thank the reviewer for his/her professional suggestion.

Fig. 4b has been replotted to show the real-time change of output (C_1 , C_2 , C_3 and C_4).

Fig. 4b The real-time output charge of four piezoelectric capacitors, from which the knock directions can be determined. Here, the positive and negative peaks of C_1 , C_2 , C_3 and C_4 are labelled with binary values of 1 and 0. Thus, the decimal value of C_1, C_2, C_3 and C_4 can be used to present the knock directions shown in the table.

Under an applied normal force or shear force in the x- or y-axis, as shown in Supplementary Fig. 10, the output charges (Q_1 , Q_2 , Q_3 and Q_4 , which are the output charges of C_1 , C_2 , C_3 and C_4 , respectively) are sensitive to the direction of applied force, which enables it to decouple the force using four piezoelectric capacitances (C_1 , C_2 , C_3 and C_4). As shown in Supplementary Fig. 21, the slipping directions can be identified successfully using the RSHTS installed on a robotic arm for touch sensing.

Supplementary Fig. 10 | Real-time output charges (Q_1 , Q_2 , Q_3 , Q_4) of four piezoelectric capacitors (C_1 , C_2 , C_3 , and C_4) under the sinusoidal forces ($f = 5$ Hz) in the x, y, and z axes. **a** Normal force in z-axis with an amplitude of 4.07 N. **b** Shear force in x-axis with an amplitude of 1.43 N. **c** Shear force in the y-axis with an amplitude of 1.44 N.

Supplementary Fig. 21 | Real-time output charges of four piezoelectric capacitors (C_1 , C_2 , C_3 , and C_4) as the RSHTS installed on a robotic arm touching an epoxy shell to identify slipping directions.

Theoretically, the proposed RSHTS with one tactile unit and a design that features the dome-shape top layer and four-circle-shaped Al electrodes in PVDF-made sensory layer can recognize the force magnitude and direction using Q_1 , Q_2 , Q_3 and Q_4 . The applied three-axis force F on the dome can be decomposed into one normal force component (F_z) and two shear force components (F_x and F_y) perpendicular to each other (Fig. R3).

Fig. R3 Reference coordinate in one tactile unit.

We assume that the force-transmission coefficients of F_x , F_y and F_z on four piezoelectric capacitors (C_1 , C_2 , C_3 and C_4) are μ_x , μ_y and μ_z , respectively, and that each piezoelectric capacitor has the same relationship between the force F and output charge Q , which is expressed as $Q = \mu F$. Thus, the relationship between four piezoelectric capacitor charge (Q_1 , Q_2 , Q_3 and Q_4) and applied force (F_x , F_y and F_z) can be expressed by

$$\begin{cases} Q_1 = Q_{1x} + Q_{1y} + Q_{1z} \\ Q_2 = Q_{2x} + Q_{2y} + Q_{2z} \\ Q_3 = Q_{3x} + Q_{3y} + Q_{3z} \\ Q_4 = Q_{4x} + Q_{4y} + Q_{4z} \end{cases} \quad (1)$$

where

$$\begin{cases} Q_{1x} = \mu_x \mu F_x \\ Q_{2x} = -\mu_x \mu F_x \\ Q_{3x} = \mu_x \mu F_x \\ Q_{4x} = -\mu_x \mu F_x \end{cases} \quad (2)$$

$$\begin{cases} Q_{1y} = -\mu_y \mu F_y \\ Q_{2y} = -\mu_y \mu F_y \\ Q_{3y} = \mu_y \mu F_y \\ Q_{4y} = \mu_y \mu F_y \end{cases} \quad (3)$$

$$\begin{cases} Q_{1z} = \mu_z \mu F_z \\ Q_{2z} = \mu_z \mu F_z \\ Q_{3z} = \mu_z \mu F_z \\ Q_{4z} = \mu_z \mu F_z \end{cases} \quad (4)$$

Substituting Eqs. (2)–(4) into Eq. (1), we can obtain

$$\begin{cases} Q_1 = \mu_x \mu F_x - \mu_y \mu F_y + \mu_z \mu F_z \\ Q_2 = -\mu_x \mu F_x - \mu_y \mu F_y + \mu_z \mu F_z \\ Q_3 = \mu_x \mu F_x + \mu_y \mu F_y + \mu_z \mu F_z \\ Q_4 = -\mu_x \mu F_x + \mu_y \mu F_y + \mu_z \mu F_z \end{cases} \quad (5)$$

Based on Eq. (5), F_x , F_y and F_z of the applied force F can be calculated via the following equation:

$$\begin{cases} F_x = k_1 Q_x \\ F_y = k_2 Q_y \\ F_z = k_3 Q_z \end{cases} \quad (6)$$

Defining

$$\begin{cases} k_1 = \frac{1}{2\mu_x \mu} \\ k_2 = \frac{1}{2\mu_y \mu} \\ k_3 = \frac{1}{\mu_z \mu} \end{cases} \quad (7)$$

$$\begin{cases} Q_x = \frac{(Q_1 + Q_3) - (Q_2 + Q_4)}{2} \\ Q_y = \frac{(Q_3 + Q_4) - (Q_1 + Q_2)}{2} \\ Q_z = \frac{Q_1 + Q_2 + Q_3 + Q_4}{4} \end{cases} \quad (8)$$

Thus, the sensitivities of the proposed sensor in three directions can be expressed by:

$$\begin{cases} Q_x = S_x F_x \\ Q_y = S_y F_y \\ Q_z = S_z F_z \end{cases} \quad (9)$$

The experiments about the sensitivities (S_x , S_y and S_z) of the RSHTS under different frequencies (5 – 600 Hz) have been carried out, and the corresponding results are shown below (Figs. R4–6). Here, we take a force in the xy-plane with a frequency range of 5 – 100 Hz in different directions applied on the sensor as examples. The magnitude and direction of applied force can be recognized, as shown in Tables R1–3.

From Tables R1–3, we can see that there are some errors in determining force magnitude and direction using Eq. (9). The reason is that the force decoupling model was established based on the assumption that C_1 , C_2 , C_3 and C_4 have the same force response, while from Supplementary Fig. 10 and Fig. R7, we can see that the actual output charges (Q_1 , Q_2 , Q_3 and Q_4) of the four piezoelectric capacitors (C_1 , C_2 , C_3 and C_4 , respectively) are not the same in terms of sensitivity due to the fabrication error. Moreover, the sensitivities of C_1 , C_2 , C_3 and C_4 are frequency-dependent (Figs. R4–6 and R8), and current force decoupling experiments are carried out under a periodic excitation with known frequency. At the same time, in real applications, the frequency of applied force is unknown. Hence, this phenomenon must be studied further in the future for real-time force decoupling.

Overall, the supplementary experimental results suggest that the RSHTS can be used to decouple the magnitude and direction of the applied force. In this work, we develop a new design for a piezoelectric tactile sensor to trigger a significantly magnified effect in a new working mode (d_{31}) of the piezoelectric sensory layer with superior sensitivity, with the goal of improving the sensitivity of the piezoelectric tactile sensor under high-frequency dynamic stimuli. We mainly focused on the demonstration of such a new design for sensitivity improvement. Meanwhile, the accurate and real-time decoupling of the direction and magnitude of arbitrary excitation must also be investigated systematically in the future. Thus, we only added related experimental results (Supplementary Figs. 10 and 21) in the supporting information.

Fig. R4 Frequency response of Q_z under normal force in the z-axis. Output charge of the RSHTS as a function of applied normal force measured within 5–600 Hz.

Fig. R5 Frequency response of Q_y under shear force in the y-axis. Output charge of the RSHTS as a function of applied shear force measured within 5–600 Hz

Fig. R6 Frequency response of Q_x under shear force in the x-axis. Output charge of the RSHTS as a function of applied shear force measured within 5–600 Hz

Table R1 The decoupled magnitude and direction of an applied force in the xy-plane with a frequency of 5 Hz.

Applied force frequency	Direction of applied force (°)	0	30	45	60	75	Averaged error
5 Hz	Calculated value of direction (°)	0	26	40	54	73	7%
	Error	0%	13%	11%	10%	3%	
$\begin{cases} Q_x = 206.6F_x \\ Q_y = 662.4F_y \\ Q_z = 260.2F_z \end{cases}$	Amplitude of applied force (N)	1.0	1.0	1.0	1.0	1.0	Averaged error
	Calculated value of amplitude (N)	1.4	1.3	1.3	1.1	0.96	23%
	Error	40%	30%	30%	10%	4%	

Table R2 The decoupled magnitude and direction of an applied force in the xy-plane with a frequency of 30 Hz.

Applied force frequency	Direction of applied force (°)	0	30	45	60	75	Averaged error
30 Hz	Calculated value of direction (°)	0	26	37	53	73	9%

$\begin{cases} Q_x = 199.2F_x \\ Q_y = 589.4F_y \\ Q_z = 306.4F_z \end{cases}$	Error	0%	13%	18%	12%	3%	
	Amplitude of applied force (N)	1.5	1.5	1.5	1.5	1.5	Averaged error
	Calculated value of amplitude (N)	1.9	1.73	1.64	1.51	1.35	
	Error	27%	15%	9%	1%	10%	12%

Table R3 The decoupled magnitude and direction of an applied force in xy-plane with a frequency of 100 Hz.

Applied force frequency	Direction of applied force (°)	0	30	45	60	75	Averaged error
100 Hz	Calculated value of direction (°)	0	26	30	46	66	16%
	Error	0%	13%	33%	23%	12%	
	Amplitude of applied force (N)	1.4	1.4	1.4	1.4	1.4	Averaged error
	Calculated value of amplitude (N)	1.6	1.5	1.34	1.1	0.84	
	Error	14%	7%	4%	21%	40%	17%

Fig. R7 Output charges of four piezoelectric capacitors (C_1 , C_2 , C_3 and C_4) as functions of applied normal and shear forces measured at 100 Hz.

Fig. R8 Frequency response of the RSHTS. Sensitivity of the RSHTS (C_2) as a function of excitation frequency from 5–600 Hz under normal force in the z-axis and shear forces in the x, y-axis.

Minor Comments:

1. “To mimic the diverse mechanical sensory function of finger, commonly used approaches for converting tactile information into electrical signals include piezoresistive, capacitive, piezoelectric, and triboelectric types.” This statement suggests the aforementioned sensors aimed to mimic the mechanical sensory function of the finger, although this might not be the motivation for the design of these sensors. The author may

consider rewriting this sentence.

Response: Thank you for the suggestion. The corresponding sentences have been rewritten in the second paragraph of the ‘Introduction’ (Page 1), as shown below:

‘To achieve the perception of mechanical force sensing for intelligent robotics and wearable electronics, tactile sensors based on piezoresistive, capacitive, triboelectric and piezoelectric mechanisms are commonly used to convert tactile information into electrical signals.’

2. “Their form often takes a layered structure of two flexible substrates sandwiching a sensory layer, where one of the flexible substrates is used to contact stimuli directly and transfers force from outer to inner sensory layer” This statement is not complete as some soft sensor uses soft and flexible substrate to allow the sensor to conform to curved surfaces [2] since some application involves interacting with a human, not just to transmit the stimuli to the sensory layer.

Response: Thank you for the suggestion. We have corrected these sentences as follows:

‘And its form often takes a layered structure composed of flexible substrates and sensory layer, wherein flexible substrates are used to contact stimuli directly and transfer force from the outer to inner sensory layer, or to conform to the curved surfaces of the robot’s body for interacting with the environment.’

3. What is the supplier of the PVDF layer is used for the fabrication of the tactile sensor? If this layer is fabricated by the authors, please explain the fabrication process.

Response: We thank the reviewer for the kind reminder. The supplier of the PVDF layer is Jinzhou Kexin Electronic Materials Co., Ltd. The corresponding information has been added in the ‘Method part (Page 13), as shown below:

Jinzhou Kexin Electronic Materials Co., Ltd, China

References

- [1] Vogt DM, Park Y-L, Wood RJ. Design and Characterization of a Soft Multi-Axis Force Sensor Using Embedded Microfluidic Channels. IEEE Sens J 2013; 13: 4056–4064.
- [2] Dahiya RS, Mittendorfer P, Valle M, et al. Directions toward effective utilization of tactile skin: A review. IEEE Sens J 2013; 13: 4121–4138.

Reviewer #2

This manuscript is on the investigation of a novel flexible tactile sensor working at high frequency for wearable electronics and intelligent robotics applications. The authors got their inspiration from our human finger’s structures, rigid skeleton embedded inside flexible muscle, to design a unique rigid-flexible coupled tactile sensor structure to achieve superior sensitivity. The authors also carried out extensive experiments and the results show that 17 times sensitivity enhancement can be achieved compared with the theoretical limit value in d_{33} mode. The physics behind such high performance is attributed to 1) enhanced transmission of high frequency dynamics force as well as 2) magnified effect of piezo-electric sensory layer in d_{31} mode. This piece of research work is very interesting and impressive with highly potential applications. Such design to achieve super sensor performance can be a very good reference for future sensor designs and researches.

I would like to recommend the Minor Revision of the manuscript before it can be accepted for publication. The authors need to address the following issue properly:

Response: We thank the reviewer for highlighting the two novel behaviours of our sensor and for recognizing the potential for intelligent robotics and wearable electronics applications. We now address the reviewer's concerns point-by-point, as shown below.

1. There are some English grammar errors inside the manuscript. The authors should correct them carefully. I scanned the reviewed copy and upload the files for their reference.

Response: Thank you very much for the excellent suggestion. We have corrected the English grammar errors in the whole manuscript according to the reviewer's detailed comments.

2. Could such structure' performance be further improved by using other flexible materials as well as optimized rigid array structures?

Response: Yes. As we have mentioned in our response 2 to Reviewer #1, a simulation study has been conducted to investigate the effects of parameters, such as the radius and height of the rigid pillar as well as the thickness and stiffness of the flexible bottom substrate, on the performance of the RSHTS (Supplementary Fig. 9). The results suggest that the sensor's sensitivity could be improved further by optimizing the parameters of the pillar and bottom substrate. The corresponding results have been added in the Supporting Information section.

3. Since the rigid structures are embedded inside flexible layers, does it form cracks after many runs of testing, typically the interfaces between flexible layer and rigid surface are the places where the mechanical stress/cracks can be formed, and such cracks grow with many runs of operating. It finally affects the structure' reliability.

Response: We thank the reviewer for the suggestion. Currently, once the RSHTS excited over 1.6×10^6 vibration cycles, no obvious mechanical stress/cracks are formed between the PDMS flexible layer and the Epoxy rigid surface (Fig. R9). This may be due to the design of our sensor, which uses a rigid structure in combination with a soft bottom substrate wherein the soft bottom substrate acts as a buffer to avoid the formation of cracks in the sensor.

Fig. R9 Photos of the rigid-soft hybrid top layer of the RSHTS. (a) Before the test. (b) After a cycling test higher than 1.6×10^6 under a normal force with an amplitude of 4.1 N.

4. Supplementary Fig. 7a, Z axis direction needs to be corrected.

Response: Thank you for the suggestion. Supplementary Fig. 11a (previously Supplementary Fig. 7a in the Supporting Information section) has been corrected as follows:

Supplementary Fig. 11 | Comparison of the signal direction detection between the RSHTS and control sensor (without rigid pillars). **a** Schematic of the finger's six slipping directions. The sensor array is attached on a 3D micro stage to simulate the finger slipping by controlling the x, y, and z axes. **b** Taking one sensory unit as an example, four piezoelectric capacitor output charges of the RSHTS (left) and the control sensor (right) are recorded in real time, respectively.

REVIEWERS' COMMENTS

Reviewer #2 (Remarks to the Author):

The authors have addressed my questions properly and the revised manuscript can be accepted for publications.